# MIM4DD: Mutual Information Maximization for Dataset Distillation

**Yuzhang Shang**[1], **Zhihang Yuan**[2], **Yan Yan**[1]*
[1]Department of Computer Science, Illinois Institute of Technology
[2]Huomo AI
`yshang4@hawk.iit.edu, zhihang.yuan@huomo.ai, yyan34@iit.edu`

## Abstract

Dataset distillation (DD) aims to synthesize a small dataset whose test performance is comparable to a full dataset using the same model. State-of-the-art (SoTA) methods optimize synthetic datasets primarily by matching heuristic indicators extracted from two networks: one from real data and one from synthetic data (see Fig. 1, Left), such as gradients and training trajectories. DD is essentially a compression problem that emphasizes maximizing the preservation of information contained in the data. We argue that well-defined metrics which measure the amount of shared information between variables in information theory are necessary for success measurement but are never considered by previous works. Thus, we introduce mutual information (MI) as the metric to quantify the shared information between the synthetic and the real datasets, and devise `MIM4DD` numerically maximizing the MI via a newly designed optimizable objective within a contrastive learning framework to update the synthetic dataset. Specifically, we designate the samples in different datasets that share the same labels as positive pairs and *vice versa* negative pairs. Then we respectively pull and push those samples in positive and negative pairs into contrastive space via minimizing NCE loss. As a result, the targeted MI can be transformed into a lower bound represented by feature maps of samples, which is numerically feasible. Experiment results show that `MIM4DD` can be implemented as an add-on module to existing SoTA DD methods.

## 1 Introduction

Deep learning has remarkably successful performance in computer vision tasks [25, 6], but most deep-learning-based methods require enormous amounts of data followed by extensive training resources. For example, in order to train a CLIP model [31] in a self-supervised manner [8], a high-resolution dataset with more than 10 million images are collected for training, which consumes tens of hundreds of GPU hours.

A straightforward solution to eliminate the reliance on training DNNs on data is to construct small training sets. Before the deep learning era, coreset or subset selection is the most prevalent paradigm, in which one can obtain a subset of salient data points to represent the original dataset of interest [1, 12, 34]. In the era of deep learning, in contrast to

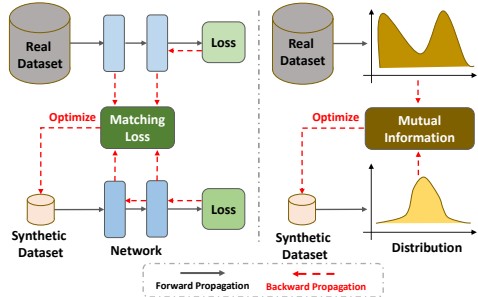

Figure 1: **(Left)** General framework of previous SoTA DD methods: matching heuristic indicators extracted by networks from real and synthetic datasets; **(Right)** the motivation of `MIM4DD`: maximizing the mutual information between two datasets for optimizing the synthetic dataset.

---

*Corresponding author

37th Conference on Neural Information Processing Systems (NeurIPS 2023).

previously mentioned selection-based methods, Dataset Distillation (DD), also known as Dataset Condensation [41, 46, 40, 5], offers a revolutionary paradigm to synthesize a small dataset using gradient descent algorithms [33], as shown in Fig.1 (Left). In DD, the key question is how to define metrics to measure the distance between synthetic and real datasets. Only by optimizing a well designed distance metric can gradient descent be properly applied to update synthetic data. To answer this question, researchers design several distance algorithms. For example, Zhao *et al.* [46] measure distance between the batch gradients of the synthetic samples and original ones; Wang *et al.* [40] use the similarity between the feature maps extracted by networks on different datasets; Cazenavette *et al.* [5] resort to the MSE between the training trajectories of networks on real and synthetic datasets.

*Dataset distillation task is essentially a compression problem with a strong emphasis on maximizing the preservation of information contained in the data* [41, 19, 11]. Admittedly, previous heuristic-designed distance metrics have achieved promising performance. However, we argue that previous works neglect the well-defined distributional metric in information theory, which is necessary for success measurement. Specifically, if we define a dataset's samples as variables, it is imperative that the high-level distributional properties of these variables from synthetic and real datasets (*e.g.*, correlations and dependencies between two datasets) should be captured and utilized to guide the update of synthetic datasets. Motivated by this notion, we introduce Mutual Information (MI), a well-formulated metric in information theory for dataset distillation. In detail, MI quantifies the information amount shared by the real and synthetic datasets. In contrast to the aforementioned works focusing on aligning the indicators extracted by different neural networks, MI can naturally capture non-linear statistical dependencies between variables and be used as a measure of true dependence, which is important information in data compression [4, 39, 36].

Based on MI metric, we propose a novel method, termed Mutual Information Maximization for Dataset Distillation (`MIM4DD`). In particular, we first formulate DD as a problem of MI maximization between two datasets. Then, we derive a numerically feasible lower bound and maximize this lower bound via contrastive learning [15, 18, 8]. Finally, we design a highly effective optimization strategy for the dataset distillation task using contrastive estimation for MI maximization. In this way, contrastive learning theoretically leads to the targeted MI maximization and also contributes to the inter-class decorrelation of synthetic samples. In other words, `MIM4DD` is built upon a contrastive learning framework to synthesize the small dataset, where samples from the synthetic dataset are pulled closer to the counterparts with the identical label in the real dataset, and pulled further away from the ones with different labels in the contrastive space. To the best of our knowledge, it is the first work aiming at MIM of the datasets over the DD task within a contrastive learning framework.

Overall, the contributions of this paper are three-fold: **(i)** To distill information from a large real dataset to a small synthetic dataset under a well-defined metric, we formulate the DD task as an MI maximization problem. To the best of our knowledge, this is the first work to introduce MI into the literature of DD. **(ii)** To maximize the targeted MI, we derive a numerically feasible lower bound and maximize it via contrastive learning. In this way, the heterogeneity in the generated images gets enhanced, which further improves the performance of the DD method. **(iii)** Experimental results show that our method outperforms existing SoTA methods. Importantly, our method can be implemented as a plug-and-play module for existing methods.

## 2 Methodology

In this section, we demonstrate the methodology of Mutual Information Maximization for Dataset Distillation (`MIM4DD`). Firstly, we briefly revisit the general framework in previous SoTA DD methods, and we illustrate the MI background. Secondly, we model the DD problem within the literature on mutual information maximization (MIM). Then, we convert the targeted but numerical inaccessible MIM goal into a learnable object optimization problem, and solve the problem via a newly designed `MIM4DD` loss. Finally, we discuss the potential insights of `MIM4DD`. Note that we only elaborate on the key derivations in this section due to the space limitation; detailed discussions and technical theorems can be found in the supplemental material.

### 2.1 Preliminaries

**Dataset Distillation.** In short, the goal of dataset distillation (also dubbed dataset condensation) is to synthesize a small training set, $\mathcal{D}_{syn} = \{\mathbf{x}_{syn}^j, \mathbf{y}^j\}_{j=1}^M$ such that models trained on this synthetic

dataset can have comparable performance as models (with the same architecture) trained on the large real set, $\mathcal{D}_{real} = \{\mathbf{x}_{real}^i, \mathbf{y}^i\}_{i=1}^N$, in which $M \ll N$. In this way, after obtaining the targeted synthetic data, the training process of DNNs can be largely accelerated. In this premise, we review some representative DD methods [43, 5, 40], which generally propose to enforce the behaviors of models trained on real and synthetic datasets to be similar. The core idea can be formalized in a form of an optimization problem:

$$\mathcal{D}_{syn}^\star = \arg \min_{\mathcal{D}_{syn}} \mathbb{E}_{\mathbf{x}_{syn} \backsim \mathcal{D}_{syn}} \mathbb{E}_{\mathbf{x}_r \backsim \mathcal{D}_{real}} \text{Dist}(f(\mathbf{x}_{real}; \theta^\star), f(\mathbf{x}_{syn}; \gamma^\star)), \qquad (2.1)$$

in which $\theta^\star$ and $\gamma^\star$ are the learned parameters of networks trained on the real dataset $\mathcal{D}_{real}$ and the synthetic dataset $\mathcal{D}_{syn}$, respectively; and the distance function $\text{Dist}(\cdot, \cdot)$ is specifically developed to measure the similarity of two networks across datasets. With aligning the networks, one can optimize the synthetic data. Note that designing the distance functions aligning the networks trained on different datasets to optimize the synthetic dataset is the core of previous DD methods, *e.g.*, CAFE [40] uses the MSE distance of feature maps, and MTT [5] calculates the difference of training trajectories as distance functions.

Although those heuristically designed distance functions lead to promising results, we argue that well-defined metrics in information theory for measuring the amount of shared information between variables have never been considered, as DD is essentially a compression problem with a different emphasis on the information contained in the data.

**Mutual Information and Contrastive Learning.** Mutual information quantifies the amount of information obtained about one random variable by observing the other random variable. It is a dimensionless quantity with (generally) units of bits, and can be considered as the reduction in uncertainty about one random variable given knowledge of another. High mutual information indicates a large reduction in uncertainty and *vice versa* [23]. Strictly, for two discrete variables $\mathbf{X}$ and $\mathbf{Y}$, their mutual information (MI) can be defined as [23]:

$$I(\mathbf{X}, \mathbf{Y}) = \sum_{x,y} P_{\mathbf{XY}}(x, y) \log \frac{P_{\mathbf{XY}}(x, y)}{P_{\mathbf{X}}(x) P_{\mathbf{Y}}(y)}, \qquad (2.2)$$

where $P_{\mathbf{XY}}(x, y)$ is the joint distribution, $P_{\mathbf{X}}(x) = \sum_y P_{\mathbf{XY}}(x, y)$ and $P_{\mathbf{Y}}(y) = \sum_x P_{\mathbf{XY}}(x, y)$ are the marginals of $\mathbf{X}$ and $\mathbf{Y}$, respectively.

In the content of DD, we would like them to share as much information as possible. Theoretically, considering the samples in real and synthetic datasets as two random variables, the MI between those two variables should be maximized. Recently, contrastive learning has been proven an effective approach to maximize MI. Many methods based on contrastive loss for self-supervised learning are proposed, such as [18], [29], [42]. These methods are theoretically based on NCE [15] and InfoNCE [18]. Essentially, the key concept of contrastive learning is to pull representations in positive pairs close and push representations in negative pairs apart in a contrastive space. Thus the major obstacle for modeling problems in a contrastive way is to define the negative and positive pairs. In this work, we also resort to a contrastive learning framework for maximizing our targeted MI. Meanwhile, we illustrate how we formulate DD as a MI maximization problem, and how we solve this targeted problem within the contrastive learning framework.

## 2.2 MIM4DD: Mutual Information Maximization for Dataset Distillation

In this section, we first formalize the idea of maximizing the MI between the distributions of real and synthesized data, via constructing a contrastive task based on Noise-Contrastive Estimation (NCE). Specifically, we derive a novel MIM4DD loss to distill task-specific information from real data $\mathcal{D}_{real}$ to synthetic data $\mathcal{D}_{syn}$, where NCE is introduced to avoid the direct MI computation by estimating it with its lower bound in Eq.2.14. From the perspective of NCE, straight-forwardly, the real and synthesized samples from the same class can be pulled close, and samples from different classes can be pushed away, which corresponds to the core idea of contrastive learning.

**Problem Formulation:** Ideally, for variable $\mathbf{X}_{real}$ representing the samples in real data and $\mathbf{X}_{syn}$ in the synthetic data, we desire to maximize the MI between $\mathbf{X}_{real}$ and $\mathbf{X}_{syn}$ in terms of $\mathbf{X}_{syn}$, *i.e.*,

$$\text{Targeted MI:} \quad \mathbf{X}_{syn}^\star = \arg \max_{\mathbf{X}_{syn}} I(\mathbf{X}_{real}, \mathbf{X}_{syn}). \qquad (2.3)$$

In this way, synthetic data $\mathcal{D}^{\star}_{syn}$ and fixed real data $\mathcal{D}^{\star}_{real}$ can share maximal information. However, directly approaching this goal is unattainable since the distributions of datasets themselves are absurd to estimate [15, 18]. To encounter this obstacle, let us include the networks trained on real and synthetic datasets. We define those networks in the form of $K$-layer Multi-Layer Perceptrons (MLPs). For simplification, we discard the bias term of those MLPs. Then the network $f(\mathbf{x})$ can be denoted as:

$$f(\mathbf{W}^1, \cdots, \mathbf{W}^K; \mathbf{x}) = (\mathbf{W}^K \cdot \sigma \cdot \mathbf{W}^{K-1} \cdots \sigma \cdot \mathbf{W}^1)(\mathbf{x}), \tag{2.4}$$

where $\mathbf{x}$ is the input sample and $\mathbf{W}^k : \mathbb{R}^{d_{k-1}} \longmapsto \mathbb{R}^{d_k} (k = 1, ..., K)$ stands for the weight matrix connecting the $(k-1)$-th and the $k$-th layer, with $d_{k-1}$ and $d_k$ representing the sizes of the input and output of the $k$-th network layer, respectively. The $\sigma(\cdot)$ function performs element-wise activation operation on the input feature maps. Based on those predefined notions, the sectional MLP $f^k(\mathbf{x})$ with the front $k$ layers of the $f(\mathbf{x})$ can be represented as:

$$f^k(\mathbf{W}^1, \cdots, \mathbf{W}^k; \mathbf{x}) = (\mathbf{W}^k \cdot \sigma \cdots \sigma \cdot \mathbf{W}^1)(\mathbf{x}). \tag{2.5}$$

The $k$-th layer's feature maps are $\mathbf{A}^k_{syn}$ and $\mathbf{A}^k_{real}, (k \in \{1, \cdots, K\})$, where $\mathbf{A}^k_{syn} = (\mathbf{a}^{k,1}_{syn}, \cdots, \mathbf{a}^{k,M}_{syn})$ and $\mathbf{A}^k_{real} = (\mathbf{a}^{k,1}_{real}, \cdots, \mathbf{a}^{k,N}_{real})$ can be considered as a series of variables. Specifically, the feature map can be obtained by:

$$\mathbf{a}^{k,j_{c_r}}_{syn} = f^k(\mathbf{x}^j_{syn}), \ j \in \{1, \cdots, M\}, \quad \mathbf{a}^{k,i_{c_s}}_{real} = f^k(\mathbf{x}^i_{real}), \ i \in \{1, \cdots, N\}. \tag{2.6}$$

Here, we show the relationship of MI in data $\mathbf{X}$ level and in feature maps $\mathbf{A}$ level, *i.e.*, the relationship between $I(\mathbf{X}_{real}, \mathbf{X}_{syn})$ and $I(\mathbf{A}_{real}, \mathbf{A}_{syn})$. We utilize the in-variance of MI to understand it. The property can be illustrated as follows:

**Theorem 1 (In-variance of Mutual Information):** Mutual information is invariant under the reparametrization of the marginal variables. If $X' = F(X)$ and $Y' = G(Y)$ are homeomorphisms (*i.e.*, $F(\cdot)$ and $G(\cdot)$ are smooth uniquely invertible maps), then

$$I(X, Y) = I(X', Y'). \tag{2.7}$$

Since each layer's mapping $\mathbf{W}^k : \mathbb{R}^{d_{k-1}} \longmapsto \mathbb{R}^{d_k} (k = 1, ..., K)$ can be considered as the smooth uniquely invertible maps **Theorem 1**. Combining this theorem with the definition of MI in Eq.2.2, we observe that the MI in the targeted data level is equivalent to MI in the feature maps level, *i.e.*,

$$I(\mathbf{X}_{real}, \mathbf{X}_{syn}) = I(\mathbf{A}^k_{real}, \mathbf{A}^k_{syn}), (k = 1, ..., K). \tag{2.8}$$

More details and the proof of this theorem are in Supplemental Materials and [21].

Facilitated by this property, we are able to access the calculation of MI between real and synthetic datasets via the feature maps of networks trained on those datasets. In other words, Theorem 1 helps us transfer the targeted MI maximization in Eq.2.3 to a reachable MI in Eq.2.8 in the dataset distillation literature, *i.e.*,

$$\text{Accessible MI:} \quad \arg\max_{\mathbf{X}_{syn}} \sum_{k=1}^{K} I(\mathbf{A}^k_{real}, \mathbf{A}^k_{syn}), \tag{2.9}$$

in which $\mathbf{A}^k_{syn} = f^k(\mathbf{X}_{syn})$ and $\mathbf{A}^k_{real} = f^k(\mathbf{X}_{real})$.

Apart from the theoretical derivation, intuitively, the corresponding variables $(\mathbf{a}^k_{syn}, \mathbf{a}^k_{real})$ should share more information, *i.e.*, MI of the same layer's output feature maps $I(\mathbf{a}^k_{syn}, \mathbf{a}^k_{real}) (k \in \{1, \cdots, K\})$ should be maximized to enforce them mutually dependent. This motivation has also been testified from the perspective of KD [17, 13] by CRD [39] and WCoRD [7]. In those methods, the penultimate layer's feature maps of teacher and student are aligned in a contrastive learning manner to enhance the heterogeneity of representations, which can also be explained via MI maximization. However, MIM4DD is different from those methods (details is in Related Work).

To optimize the accessible Mutual Information (MI) as defined in Eq.2.9, we incorporate a contrastive learning framework into our targeted Dataset Distillation (DD) task. The fundamental principle of contrastive learning involves comparing varying perspectives of the data, typically under different data augmentations, to compute similarity scores [29, 18, 2, 16, 8]. This framework is suitable for our case, since the activations from real and synthetic datasets can be seen as two different views.

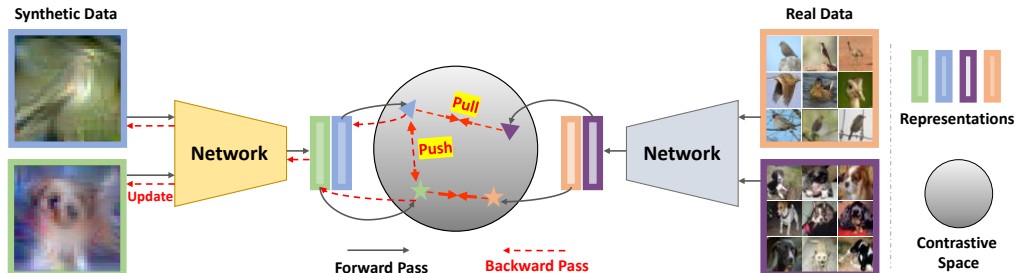

Figure 2: Feeding images from two datasets into two corresponding neural networks, and we obtain the three pairs of representations for each layer. We embed the representations into a contrastive space, then learn from the pair correlation with the contrastive learning task in Eq. 2.16. In this way, the heterogeneity in the generated images can be enhanced. Theoretically, we can realize the formulated mutual information maximization (MIM) in Eq. 2.9, which is equivalent to the targeted MIM in Eq. 2.3.

**(Definition: Positive and Negative Samples.)** Let's denote each sample from the real and synthetic datasets as $\{\mathbf{x}_{real}^{j_{c_r}}\}$ $(j_{c_r} \in \{1, \cdots, N\})$ and $\{\mathbf{x}_{syn}^{i_{c_s}}\}$ $(i_{c_s} \in \{1, \cdots, M\})$, respectively. Here, $c_r$ and $c_s \in \{1, \cdots, C\}$ represent the class labels of $\mathbf{x}_{real}^{j_{c_r}}$ and $\mathbf{x}_{syn}^{i_{c_s}}$, respectively. We feed two batches of samples from two datasets to two different neural networks and obtain $N \cdot M$ pairs of $k$-th layer's activations $(\mathbf{a}_{syn}^{k,i_{c_s}}, \mathbf{a}_{real}^{k,j_{c_r}})$, which can further be utilized for the contrastive learning task. We define a pair containing two activations corresponding to the same labels as the positive pair, *i.e.*, if $c_r = c_s$, $(\mathbf{a}_{syn}^{k,i_{c_s}}, \mathbf{a}_{real}^{k,j_{c_r}})_+$ and *vice versa*. Consequently, there is $\frac{M}{C} \cdot \frac{M}{C} \cdot C = \frac{1}{C} \cdot N \cdot M$ positive pairs, and thus $(1 - \frac{1}{C}) \cdot N \cdot M$ negative pairs. The key concept of contrastive learning is to discriminate whether a given pair of activation $(\mathbf{a}_{syn}^{k,i_{c_s}}, \mathbf{a}_{real}^{k,j_{c_r}})$ is positive or negative. In other words, it involves estimate the distribution $P(D \mid \mathbf{a}_{syn}^{k,i_{c_s}}, \mathbf{a}_{real}^{k,j_{c_r}})$, in which $D$ is a scalar variable indicating whether $c_s = c_r$ or $c_s \neq c_r$. Specifically, $D = 1$ when $c_s = c_r$, and $D = 0$ when $c_s \neq c_r$. However, we can not directly reach the distribution $P(D \mid \mathbf{a}_{syn}^{k,i_{c_s}}, \mathbf{a}_{real}^{k,j_{c_r}})$ [15], and thus we introduce its corresponding variational form:

$$q(D \mid \mathbf{a}_{real}^{k,j_{c_r}}, \mathbf{a}_{syn}^{k,i_{c_s}}). \tag{2.10}$$

Intuitively, $q(D \mid \mathbf{a}_{real}^{k,j_{c_r}}, \mathbf{a}_{syn}^{k,i_{c_s}})$ can be treated as a binary classifier, which can classify a given pair $(\mathbf{a}_{syn}^{k,i_{c_s}}, \mathbf{a}_{real}^{k,j_{c_r}})$ into positive or negative. Importantly, $q(D \mid \mathbf{a}_{real}^{k,i_{c_s}}, \mathbf{a}_{syn}^{k,j_{c_r}})$ can be estimated by some mature statistical methods, such as NCE [15] and InfoNCE [18].

Using the Bayes rule, the posterior probability of two activations from the positive pair can be formalized as:

$$q(D = 1 \mid \mathbf{a}_{syn}^{k,i_{c_s}}, \mathbf{a}_{real}^{k,j_{c_r}}) = \frac{q(\mathbf{a}_{syn}^{k,i_{c_s}}, \mathbf{a}_{real}^{k,j_{c_r}} \mid D = 1)\frac{1}{C}}{q(\mathbf{a}_{syn}^{k,i_{c_s}}, \mathbf{a}_{real}^{k,j_{c_r}} \mid D = 1)\frac{1}{C} + q(\mathbf{a}_{syn}^{k,i_{c_s}}, \mathbf{a}_{real}^{k,j_{c_r}} \mid D = 0)\frac{C-1}{C}}. \tag{2.11}$$

The probability of activations from negative pair is $q(D = 0 \mid \mathbf{a}_{syn}^{k,i_{c_s}}, \mathbf{a}_{real}^{k,j_{c_r}}) = 1 - q(D = 1 \mid \mathbf{a}_{syn}^{k,i_{c_s}}, \mathbf{a}_{real}^{k,j_{c_r}})$. To simplify the NCE derivative, several works [15, 42, 39] build assumption about the dependence of the variables, we also use the assumption that the activations from positive pairs are dependent and the ones from negative pairs are independent, *i.e.* $q(\mathbf{a}_{syn}^{k,i_{c_s}}, \mathbf{a}_{real}^{k,j_{c_r}} \mid D = 1) = P(\mathbf{a}_{syn}^{k,i_{c_s}}, \mathbf{a}_{real}^{k,j_{c_r}})$ and $q(\mathbf{a}_{syn}^{k,i_{c_s}}, \mathbf{a}_{real}^{k,j_{c_r}} \mid D = 0) = P(\mathbf{a}_{syn}^{k,i_{c_s}})P(\mathbf{a}_{real}^{k,j_{c_r}})$. Hence, the above equation can be simplified as:

$$q(D = 1 \mid \mathbf{a}_{syn}^{k,i_{c_s}}, \mathbf{a}_{real}^{k,j_{c_r}}) = \frac{P(\mathbf{a}_{syn}^{k,i_{c_s}}, \mathbf{a}_{real}^{k,j_{c_r}})}{P(\mathbf{a}_{syn}^{k,i_{c_s}}, \mathbf{a}_{real}^{k,j_{c_r}}) + P(\mathbf{a}_{syn}^{k,i_{c_s}})P(\mathbf{a}_{real}^{k,j_{c_r}})(C-1)}. \tag{2.12}$$

Performing logarithm to Eq.2.12 and arranging the terms, we can achieve

$$\log q(D = 1 \mid \mathbf{a}_{syn}^{k,i_{c_s}}, \mathbf{a}_{real}^{k,j_{c_r}}) \leq \log \frac{P(\mathbf{a}_{syn}^{k,i_{c_s}}, \mathbf{a}_{real}^{k,j_{c_r}})}{P(\mathbf{a}_{syn}^{k,i_{c_s}})P(\mathbf{a}_{real}^{k,j_{c_r}})} - \log(C-1). \tag{2.13}$$

Taking expectation of $P(\mathbf{a}_{syn}^{k,i_{cs}}, \mathbf{a}_{real}^{k,j_{cr}})$ on both sides, and combining Eq.2.2, we can transfer the MI into:

$$\overbrace{I(\mathbf{a}_{syn}^k, \mathbf{a}_{real}^k)}^{\text{Accessible MI in Eq. 2.9}} \geq \log(C-1) + \overbrace{\mathbb{E}_{P(\mathbf{a}_{syn}^{k,i_{cs}}, \mathbf{a}_{real}^{k,j_{cr}}|D=1)}\left[\log q(D=1 \mid \mathbf{a}_{syn}^{k,i_{cs}}, \mathbf{a}_{real}^{k,j_{cr}})\right]}^{\text{optimized lower bound}}, \quad (2.14)$$

where $I(\mathbf{a}_{syn}^k, \mathbf{a}_{real}^k)$ is the MI between the real and synthetic data distributions. Instead of directly maximizing the MI, maximizing the lower bound in the Eq.2.14 is a practical solution.

However, even $q(D = 1 \mid \mathbf{a}^{k,i_{cs}}, \mathbf{a}^{k,j_{cr}})$ is still hard to estimate. Thus, as tackled by many contrastive learning works [29, 18, 2, 39, 36, 37], we introduce a discriminator network $d(\cdot, \cdot)$ with parameter $\phi$ (*i.e.*, $d(\mathbf{a}_{syn}^{k,i_{cs}}, \mathbf{a}_{real}^{k,j_{cr}}; \phi)$). Basically, the discriminator $d$ can map $\mathbf{a}_{syn}^k, \mathbf{a}_{real}^k$ to $[0, 1]$ (*i.e.*, distinguish given two samples $\mathbf{a}_{syn}^k, \mathbf{a}_{real}^k$ belonging to positive or negative pair). Specifically, the discriminator function is designed as follows:

$$d(\mathbf{a}_{syn}^{k,i_{cs}}, \mathbf{a}_{real}^{k,j_{cr}}) = \exp(\frac{< g_\phi(\mathbf{a}_{syn}^{k,i_{cs}}), g_\phi(\mathbf{a}_{real}^{k,j_{cr}}) >}{\tau})/C, \quad (2.15)$$

in which $g_\phi(\cdot)$ is the embedding function for mapping the activations into the contrastive space and $C = \exp(\frac{< g(\mathbf{a}_{syn}^{k,i_{cs}}), g(\mathbf{a}_{real}^{k,j_{cr}}) >}{\tau}) + 1$, and $\tau$ is a temperature parameter that controls the concentration level of the distribution [17, 39].

**Loss Function.** We define the contrastive loss function $\mathcal{L}_{NCE}^k$ between the $k$-th layer's activations $\mathbf{a}_{syn}^k$ and $\mathbf{a}_{real}^k$ as: $\mathcal{L}_{NCE}^k =$

$$\mathbb{E}_{q(\mathbf{a}_{syn}^{k,i_{cs}}, \mathbf{a}_{real}^{k,j_{cr}}|D=1)}\left[\log h(\mathbf{a}_{syn}^{k,i_{cs}}, \mathbf{a}_{real}^{k,j_{cr}})\right] + (C-1)\cdot\mathbb{E}_{q(\mathbf{a}_{syn}^{k,i_{cs}}, \mathbf{a}_{real}^{k,j_{cr}}|D=0)}\left[\log(1 - h(\mathbf{a}_{syn}^{k,i_{cs}}, \mathbf{a}_{real}^{k,j_{cr}}))\right].$$
$$(2.16)$$

In the view of contrastive learning, the first term in the above loss function about positive pairs is optimized for capturing more intra-class correlations and the second term of negative pairs is for inter-class decorrelation. Because we construct the pairs instance-wisely, the number of negative pairs can be the size of the entire real dataset, *e.g.*, 50K for CIFAR [22]. By incorporating hand-crafted, additional contrastive pairs for the proxy optimization problem in Eq.2.16, the representational quality of generated images can be further enhanced as demonstrated by numerous contrastive learning methods[8, 29, 18, 2].

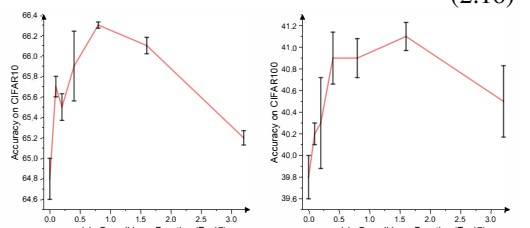

Figure 3: Ablation Studies. Effect of $\lambda$ in $\mathcal{L}_{\text{MIM4DD}}$ (Eq.2.17) on CIFAR10 (Left) and CIFAR100 (Right). $\lambda = 0$ is the baseline, MTT [5].

Finally, by incorporating the set of NCE loss for various layers $\left\{\mathcal{L}_{NCE}^k\right\}, (k = 1, \cdots, K)$, we can then formulate MIM4DD loss $\mathcal{L}_{\text{MIM4DD}}$ as:

$$\mathcal{L}_{\text{MIM4DD}} = \lambda \sum_{k=1}^{K} \frac{\mathcal{L}_{NCE}^k}{\beta^{K-1-k}} + \mathcal{L}_{DD}, \quad (2.17)$$

in which $\mathcal{L}_{DD}$ can be any loss functions in previous DD methods [40, 43, 5], $\lambda$ is used to control the NCE loss, and $\beta$ is a scalar greater than 1. In practice, we use MTT [5] or BPTT [11] as default $\mathcal{L}_{DD}$.

## 2.3 Discussion on MIM4DD

In addition to the theoretical formulation, we are going to provide a more straightforward manifestation of MIM4DD. As illustrated in Fig.2, by embedding the activations to the contrastive space and constructing proper negative-and-positive embedding pairs, we can heterogeneously distill the relevant information from the real dataset to learnable synthetic dataset, and enhance the heterogeneity of the synthetic dataset within the contrastive learning framework as shown in Fig.5. Specifically, networks on different datasets first learn self-supervised information benefited from a large number of the designed pairs [18, 39]. Then, via the backward pass in Fig.2, the synthetic data are updated

w.r.t. this contrastive loss. While the actual number of negative samples in practice can be huge, *e.g.*, 50K for synthesizing 10 pictures per-class to replace CIFAR10 [22] despite only drawing two samples in Fig.2.

Furthermore, we give a direct explanation of *why optimizing* $\mathcal{L}_{MIM4DD}$ *in* Eq.2.17 *can lead to the targeted MIM* between synthetic dataset and real dataset in Eq.2.3. Firstly, optimizing the formulated contrastive loss in Eq.2.16 is equivalent to maximizing the derived accessible MI between activations in Eq.2.9, as the inequality in Eq.2.14 (Eq.2.10-2.13 can derive Eq.2.14). Secondly, based on Theorem 1 (Eq.A.1) and the property of the networks itself (Eq.2.4 and Eq.2.5), maximizing the accessible MI in Eq.2.9 equals to maximizing the targeted MI in Eq.2.8, which is our goal, to minimize the MI between synthetic dataset and real dataset. Since real data are fixed and synthetic data are learnable, the optimized synthetic dataset can contain as much information as the real dataset under the metric of MI.

# 3 Experiments

In this section, we conduct comprehensive experiments to evaluate our proposed method `MIM4DD` on four different datasets for DD task. We first describe the implementation details of `MIM4DD`, and then compare our method with several SoTA DD methods to demonstrate superiority of our proposed method. Finally, we validate the effectiveness of MI module (connected with Eq.2.16 and Eq.2.17) by a series of ablation studies.

## 3.1 Datasets and Implementation Details

**Datasets.** We use MNIST [24], SVHN [35], and CIFAR10/100 datasets to conduct our experiments. **MNIST** [24] is a dataset for handwritten digits recognition that is widely used for validating image recognition models. It contains 60,000 training images and 10,000 testing images with the size of $28 \times 28$. **CIFAR10/100** [22] are two datasets consist of tiny colored natural images with the size of $32 \times 32$ from 10 and 100 categories, respectively. In each dataset, 50,000 images are used for training and 10,000 images for testing. More details of the datasets can be found in *Supplemental Materials*.

**Implementation Details.** In the experiments, we optimize synthetic sets with 1/10/50 Images Per Class (IPC) across all three datasets, using a three-layer Convolutional Network (ConvNet) identical to those used in [46, 40, 5]. The ConvNet comprises three consecutive blocks of 'Conv-InstNorm-ReLU-AvgPool.' Each convolutional layer has 128 channels, and AvgPool represents a $2 \times 2$ average pooling operation with stride 2. The synthetic images' initial learning rate is 0.1, which is halved at the 1,800th and 2,800th iterations. The training is stopped after 5,000 iterations. To test the ConvNet's performance on the synthetic dataset, we train the network on synthetic sets for 300 epochs and assess the performance using five randomly initialized networks. The network's initial learning rate is 0.01. As per [5], we conduct five experiments and report the mean and standard deviation across the five networks. The default batch size is 256, and $\lambda$ in Eq.2.17 is 0.8. The effect of $\lambda$ is explored in Sec.3.3.

## 3.2 Comparison with SoTA

We compare `MIM4DD` with a series of state-of-the-art (SoTA) dataset distillation methods, including Dataset Distillation (DD) [41], LD [3], Dataset Condensation (DC) [46], DC with differentiable siamese augmentation (DSA) [43], DC with distribution matching (DM) [45], CAFE [40], FRePo [47], TESLA [10], BPTT [11], and MTT [5]. We report the performances of our method and comparisons on three datasets in Table 1. Taking into account the overall performance of `MIM4DD` across these mainstream dataset distillation benchmarks, it's apparent that our method consistently outperforms existing SoTAs. For instance, in the setting of generating 10 images per class, our method delivers top-tier results across all datasets. Additionally, when we synthesize 10 images per class using CIFAR100 as a real-world dataset, our method surpasses MTT by a margin of 1.4%. Importantly, our method can serve as an effective plug-and-play module for existing state-of-the-art DD methods.

## 3.3 Ablation Studies

Table 1: Dataset distillation methods comparisons. The settings are the same as previous SoTAs, BPTT [11], MTT [5], and DREAM [27]. Importantly, `MIM4DD` can work as an add-on module for SoTA methods.

| | MNIST | | | CIFAR10 | | | CIFAR100 | |
|---|---|---|---|---|---|---|---|---|
| | IPC-1 | IPC-10 | IPC-50 | IPC-1 | IPC-10 | IPC-50 | IPC-1 | IPC-10 |
| Full Set | | $99.6 \pm 0.0$ | | | $84.8 \pm 0.1$ | | | $56.2 \pm 0.3$ |
| DD [41] | - | $79.5 \pm 8.1$ | - | - | $36.8 \pm 1.2$ | - | - | - |
| LD [3] | $60.9 \pm 3.2$ | $87.3 \pm 0.7$ | $93.3 \pm 0.3$ | $25.7 \pm 0.7$ | $38.3 \pm 0.4$ | $42.5 \pm 0.4$ | $11.5 \pm 0.4$ | $31.5 \pm 0.2$ |
| CAFE [40] | $93.1 \pm 0.3$ | $97.2 \pm 0.2$ | $98.6 \pm 0.2$ | $30.3 \pm 1.1$ | $46.3 \pm 0.6$ | $55.5 \pm 0.6$ | $14.0 \pm 0.3$ | $31.5 \pm 0.2$ |
| DM [44] | $89.7 \pm 0.6$ | $97.5 \pm 0.1$ | $98.6 \pm 0.1$ | $26.0 \pm 0.8$ | $48.9 \pm 0.6$ | $63.0 \pm 0.4$ | $11.4 \pm 0.3$ | $29.7 \pm 0.3$ |
| DSA [43] | $88.7 \pm 0.6$ | $97.8 \pm 0.1$ | $99.2 \pm 0.1$ | $28.8 \pm 0.7$ | $52.1 \pm 0.5$ | $60.6 \pm 0.5$ | $16.8 \pm 0.2$ | $32.3 \pm 0.3$ |
| DC [46] | $91.7 \pm 0.5$ | $97.4 \pm 0.2$ | $98.9 \pm 0.2$ | $28.3 \pm 0.5$ | $44.9 \pm 0.5$ | $53.9 \pm 0.5$ | $12.8 \pm 0.3$ | $25.2 \pm 0.3$ |
| DCC [26] | - | - | - | $32.9 \pm 0.8$ | $49.4 \pm 0.5$ | $61.6 \pm 0.4$ | $13.3 \pm 0.3$ | $30.6 \pm 0.4$ |
| DSAC [26] | - | - | - | $34.0 \pm 0.7$ | $54.5 \pm 0.5$ | $64.2 \pm 0.4$ | $14.6 \pm 0.3$ | $33.5 \pm 0.3$ |
| FRePo [47] | $92.4 \pm 0.5$ | $98.4 \pm 0.1$ | $98.8 \pm 0.1$ | $41.3 \pm 0.5$ | $59.6 \pm 0.3$ | $63.6 \pm 0.2$ | $24.8 \pm 0.2$ | $31.2 \pm 0.2$ |
| FRePo-w [47] | $93.0 \pm 0.4$ | $98.6 \pm 0.1$ | $99.2 \pm 0.0$ | $46.8 \pm 0.7$ | $65.5 \pm 0.4$ | $71.7 \pm 0.2$ | $28.7 \pm 0.1$ | $42.5 \pm 0.2$ |
| MTT [5] | $91.4 \pm 0.9$ | $97.3 \pm 0.1$ | $98.5 \pm 0.1$ | $46.3 \pm 0.8$ | $65.3 \pm 0.7$ | $71.6 \pm 0.2$ | $24.3 \pm 0.3$ | $40.1 \pm 0.4$ |
| TESLA [10] | - | - | - | $48.5 \pm 0.8$ | $66.4 \pm 0.8$ | $72.6 \pm 0.7$ | $24.8 \pm 0.4$ | $41.7 \pm 0.3$ |
| MTT [5] | $91.4 \pm 0.9$ | $97.3 \pm 0.1$ | $98.5 \pm 0.1$ | $46.3 \pm 0.8$ | $65.3 \pm 0.7$ | $71.6 \pm 0.2$ | $24.3 \pm 0.3$ | $40.1 \pm 0.4$ |
| + MIM4DD | $92.0 \pm 0.6$ | $98.1 \pm 0.2$ | $98.9 \pm 0.2$ | $47.6 \pm 0.2$ | $66.4 \pm 0.2$ | $71.4 \pm 0.3$ | $25.1 \pm 0.3$ | $41.5 \pm 0.2$ |
| Δ | (0.6↑) | (0.8↑) | (0.4↑) | (1.3↑) | (1.1↑) | (0.2↓) | (0.8↑) | (1.4↑) |
| BPTT [11] | $94.7 \pm 0.2$ | $98.9 \pm 0.1$ | $99.2 \pm 0.0$ | $49.1 \pm 0.6$ | $62.4 \pm 0.4$ | $70.5 \pm 0.4$ | $21.3 \pm 0.6$ | $34.7 \pm 0.5$ |
| + MIM4DD | $95.8 \pm 0.3$ | $98.9 \pm 0.1$ | $99.2 \pm 0.1$ | $51.8 \pm 0.3$ | $66.4 \pm 0.8$ | $72.9 \pm 0.5$ | $25.0 \pm 0.4$ | $38.5 \pm 0.6$ |
| Δ | (1.1↑) | (0.0-) | (0.0-) | (2.7↑) | (4.0↑) | (2.4↑) | (3.7↑) | (3.8↑) |
| DREAM [27] | - | - | - | $51.1 \pm 0.3$ | $69.4 \pm 0.4$ | $74.8 \pm 0.1$ | $29.5 \pm 0.3$ | $46.8 \pm 0.7$ |
| + MIM4DD | - | - | - | $51.9 \pm 0.3$ | $70.8 \pm 0.1$ | $74.7 \pm 0.2$ | $31.1 \pm 0.4$ | $47.4 \pm 0.3$ |
| Δ | - | - | - | (0.8↑) | (1.4↑) | (0.1↓) | (0.6↑) | (0.6↑) |

We conduct a series of ablation studies of `MIM4DD` in CIFAR-10 and CIFAR100. By adjusting the coefficient $\lambda$ in the loss function $\mathcal{L}_{\texttt{MIM4DD}}$ (Eq.2.16 and Eq.2.17), we investigate the effect of `MIM4DD` loss in synthesizing distilled datasets. The results are shown in Fig.3. We observe the trend that with $\lambda$ increasing, the performance improves, which validates the effectiveness of our designed method. However, when the ratio of $\mathcal{L}_{NCE}$ in overall loss is greater than a threshold, the performance of student networks drops, which means the main task, dataset distillation is overlooked in the optimization.

Table 2: Hyperparameters selection w. 10 Imgs/Cls on CIFAR10.

| Hyper-parameter | Accuracy |
|---|---|
| critic function w.o network | $64.9 \pm 0.2$ |
| critic function w. 1 fc-layer | $\mathbf{65.9 \pm 0.4}$ |
| critic function w. 2 fc-layer | $65.3 \pm 0.4$ |
| $\beta = 1.0$ in $\mathcal{L}_{\texttt{MIM4DD}}$ | $63.8 \pm 0.6$ |
| $\beta = 2.0$ in $\mathcal{L}_{\texttt{MIM4DD}}$ | $\mathbf{66.0 \pm 0.5}$ |
| $\beta = 0.5$ in $\mathcal{L}_{\texttt{MIM4DD}}$ | $62.8 \pm 0.6$ |

`MIM4DD` **as an Add-on Module.** We apply the `MIM4DD` framework to state-of-the-art Dataset Distillation (DD) techniques, including MTT [5] and BPTT [11]. The results are presented in Table1. It is observed that `MIM4DD` effectively enhances the performance of all the tested methods, providing substantial evidence that our approach can be utilized as an add-on module to existing techniques.

**Hyper-parameters and Relative Module Selection.** In addition to ablation studies, we conduct experiments to select the important hyper-parameters and modules. We investigate one hyper-parameter, the co-efficient to adjust the weight of layer in overall loss in Eq.2.17, and one module, the architecture of embedding network in Eq.2.15. The results are in Table 2.

### 3.4 Regularization Propriety

Here, we analyze the trends of dataset distillation loss, $\mathcal{L}_{DD}$ in Eq.2.17 during training. As we presented in Sec. 2.2, the term $\mathcal{L}_{DD}$ in Eq.2.17 can be any dataset distillation loss. By controling the co-efficient $\lambda$ in Eq.2.17, we can separately study the DD loss $\mathcal{L}_{DD}$. Their training curves are presented in Fig.4. When turning on NCE loss (assigning $\lambda = 0.8$), we observe that the $\mathcal{L}_{DD}$ drops using our designed NCE loss (Left), while the corresponding testing performance improves (see Table 1).

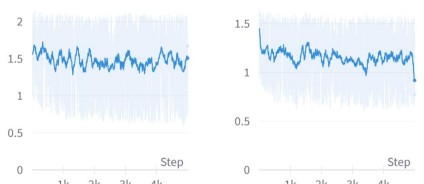

Figure 4: $\mathcal{L}_{DD}$ curves while training *w.o.* (left) and *w.* (right) $\mathcal{L}_{NCE}$.

Combining this phenomenon and analysis on the effect of $\lambda$ in Fig. 3, we conclude that our designed `MIM4DD` module can act as a regularization term.

### 3.5 CKA analysis

We formulate DD in a view of variable distribution, and thus we analyze distributional information flow within layers of neural networks. Centered kernel alignment (CKA) [9, 20, 32] is able to compare activations within or across networks quantitatively. Specifically, for a network fed by $m$ samples, CKA algorithm takes $\mathbf{X} \in \mathbb{R}^{m \times p_1}$ and $\mathbf{Y} \in \mathbb{R}^{m \times p_2}$ as inputs which are output activations of two layers (with $p_1$ and $p_2$ neurons respectively). Letting $\mathbf{K} \triangleq \mathbf{XX}^\top$ and $\mathbf{L} \triangleq \mathbf{YY}^\top$ denote the Gram matrices for the two layers CKA computes: $\mathrm{CKA}(\mathbf{K}, \mathbf{L}) = \frac{\mathrm{HSIC}(\mathbf{K},\mathbf{L})}{\sqrt{\mathrm{HSIC}(\mathbf{K},\mathbf{K})\mathrm{HSIC}(\mathbf{L},\mathbf{L})}}$,

where HSIC is the Hilbert-Schmidt independence criterion [14]. Given the centering matrix $\mathbf{H} = \mathbf{I}_n - \frac{1}{n}\mathbf{11}^\top$ and the centered Gram matrices $\mathbf{K}' = \mathbf{HKH}$ and $\mathbf{L}' = \mathbf{HLH}$, $\mathrm{HSIC} = \frac{\mathrm{vec}(\mathbf{K}')\mathrm{vec}(\mathbf{L}')}{(m-1)^2}$,

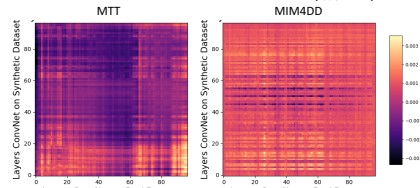

the similarity between these centered Gram matrices. Therefore, we employ Centered Kernel Alignment (CKA) to delve into the information interplay between real and synthetic data. Specifically, we input two datasets into two correspondingly trained networks and examine the similarity between the networks' feature maps using CKA. The findings are depicted in Fig.5. The CKA heatmap reveals that the dataset distilled via `MIM4DD` shares more information with the real dataset compared to the MTT [5]. This is because, in our DD formulation, the output similarity is supposed to be highly related to the data similarity (Theorem 1, Eq.A.1).

Figure 5: CKA analyzes the information shared within different datasets (Real v.s. Synthetic with 10 Img/Cls on CIFAR100). The lighter the dot, the more similar of the two corresponding layers learned from different datasets. Higher similar score between two layers' output represents those two layers share more information.

## 4 Related Work

Dataset Distillation is essentially a compression problem that emphasizes maximizing the preservation of information contained in the data. We argue that well-defined metrics which measure the amount of shared information between variables in information theory are necessary for success measurement but are never considered by previous works. Therefore, we propose introducing a well-defined metric in information theory, mutual information (MI), to guide the optimization of synthetic datasets.

The formulation of our method for DD, `MIM4DD` absorbs the core idea of contrastive learning (*i.e.*, constructing the informative positive and negative pairs for contrastive loss) of the existing contrastive learning methods, especially the contrastive KD methods, CRD [39] and WCoRD [7]. However, our approach has several differences from those methods: (i) our targeted MI and formulated numerical problem are totally different; (ii) our method can naturally avoid the cost of MemoryBank [42] for the exponential number of negative pairs in CRD and WCoRD, thanks to the small size of the synthetic dataset in our task. (Further details can be found in the Appendix.)

## 5 Conclusion

In this paper, we explore that well-defined metrics which measures the amount of shared information between variables in information theory for dataset distillation. Specifically, we introduce MI as the metric to quantify the shared information between the synthetic and the real datasets, and devise `MIM4DD` numerically maximizing the MI via a newly designed optimizable objective within a contrastive learning framework to update the synthetic dataset.

## Acknowledgements

This work is supported by NSF IIS-2309073 and NSF SCH-2123521. This article solely reflects the opinions and conclusions of its authors and not the funding agency.

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

# A Appendix

## A.1 In-variance of Mutual Information

**Theorem 1 (In-variance of Mutual Information):** Mutual information is invariant under reparametrization of the marginal variables. If $X' = F(X)$ and $Y' = G(Y)$ are homeomorphisms (*i.e.*, $F(\cdot)$ and $G(\cdot)$ are smooth uniquely invertible maps), then

$$I(X, Y) = I(X', Y'). \tag{A.1}$$

**Proof.** If $X' = F(X)$ and $Y' = G(Y)$ are homeomorphisms (smooth and uniquely invertible maps), and $J_X = \|\frac{\partial X}{\partial X'}\|$ and $J_Y = \|\frac{\partial Y}{\partial Y'}\|$ are the Jacobi determinants, then

$$\mu'(x', y') = J_X(x')J_Y(y')\mu(x, y) \tag{A.2}$$

and similarly for the marginal densities, which gives

$$
\begin{aligned}
I(X', Y') &= \iint dx'dy'\mu'(x', y') \log \frac{\mu'(x', y')}{\mu'_x(x')\mu'_y(y')} \\
&= \iint dxdy\mu(x, y) \log \frac{\mu(x, y)}{\mu_x(x)\mu_y(y)} \\
&= I(X, Y).
\end{aligned}
\tag{A.3}
$$

More details can be found in [21].

**Discussion on Theorem 1.**

Our objective is to maximize the Mutual Information (MI) between the synthetic dataset and the real dataset (Eq. 3), a task that is numerically unfeasible. To overcome this challenge, we present this theorem. It allows us to transform the target problem at the data level (Eq. 3) into a more manageable problem at the feature map level (Eq. 9). Given that each layer's mapping $\mathbf{W}^k : \mathbb{R}^{d_{k-1}} \longmapsto \mathbb{R}^{d_k} (k = 1, ..., K)$ in the network (as per Eq. 4, 5, and 6) can be treated as smooth, uniquely invertible maps, we can achieve the goal of maximizing the mutual information between the two datasets. This is done by maximizing the mutual information between two sets of down-sampled feature maps.

## A.2 Datasets and Implementation Details

### A.2.1 Datasets

**MNIST** [24] is a dataset for handwritten digits recognition that is widely used for validating image recognition models. It contains 60,000 training images and 10,000 testing images with the size of $28 \times 28$.

**CIFAR10/100** [22] are two datasets consist of tiny colored natural images with the size of $32 \times 32$ from 10 and 100 categories, respectively. In each dataset, 50,000 images are used for training and 10,000 images for testing.

### A.2.2 Implementation Details.

In the experiments, we optimize synthetic sets with 1/10/50 Images Per Class (IPC) across all three datasets, using a three-layer Convolutional Network (ConvNet) identical to those used in [46, 40, 5]. The ConvNet comprises three consecutive blocks of 'Conv-InstNorm-ReLU-AvgPool.' Each convolutional layer has 128 channels, and AvgPool represents a $2 \times 2$ average pooling operation with stride 2. The synthetic images' initial learning rate is 0.1, which is halved at the 1,800th and 2,800th iterations. The training is stopped after 5,000 iterations. To test the ConvNet's performance on the synthetic dataset, we train the network on synthetic sets for 300 epochs and assess the performance using five randomly initialized networks. The network's initial learning rate is 0.01. As per [5], we conduct five experiments and report the mean and standard deviation across the five networks. The default batch size is 256, and $\lambda$ in Eq.17 is 0.8. The effect of $\lambda$ is explored in Sec.3.3.

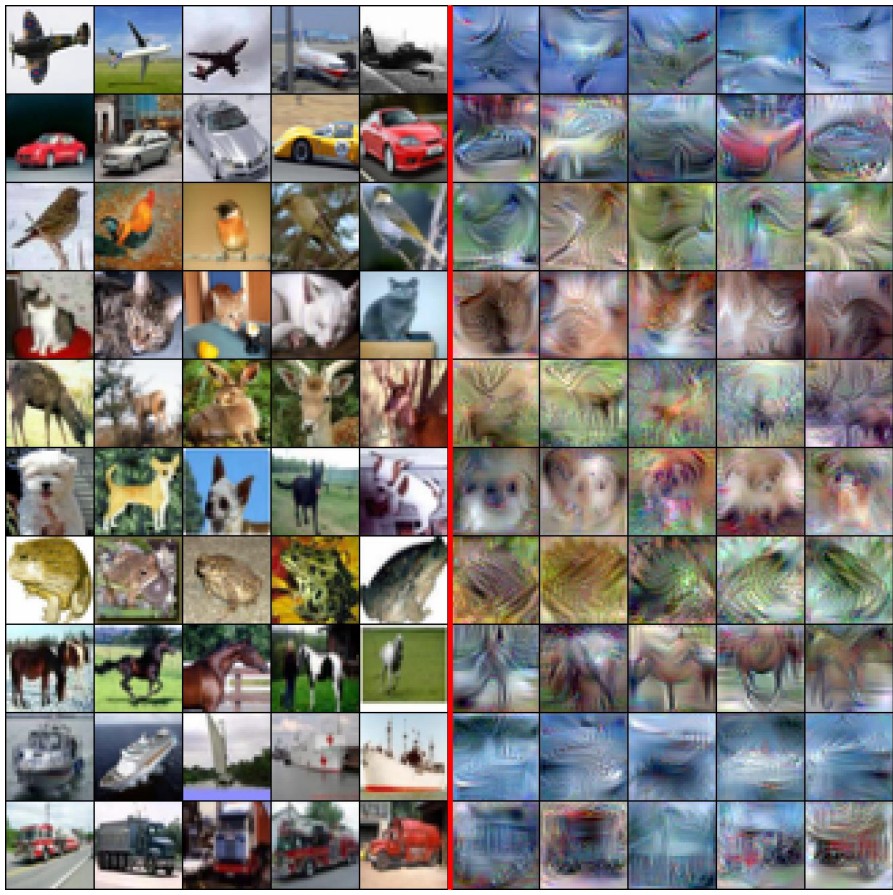

Figure 6: **(Left)** Samples from CIFAR10; **(Right)** Samples from Synthetic dataset based on CIFAR10. We observe that the heterogeneity in the generated images enhanced, benefited from the contrastive learning loss (Loss $\mathcal{L}_{\texttt{MIM4DD}}$ in Eq.17).

### A.3 Synthetic Samples Visualization.

## B  Related Work

**Dataset Distillation** (DD) is firstly introduced by Wang *et al.* [41], in which they optimize the distilled images using gradient-based hyperparameter optimization [28]. The key problem is to optimize the specific-designed metrics of networks on real and synthetic datasets to update the optimizable images. Subsequently, several works significantly improve the results by designing different metrics. For example, Bohdal *et al.*and Sucholutsky *et al.* [3, 38] use distance between networks' soft labels; Zhao *et al.* [46] define the gradients of networks as metric; Zhao *et al.* [43] further adopts augmentations to enhance the alignment ability; Wang *et al.* [40] utilize distance of network feature maps as metric; and Cazenavette [5] propose long-range trajectory to construct the metric function. Lee *et al.* [26] propose Dataset Condensation with Contrastive Signals (DCC) by modifying the loss function to enable the DC methods to effectively capture the differences between classes. On the other hand, researchers take DD as a bi-level optimization problem. For example, Zhou *et al.* [47] employ a closed-form approximation for the unrolled inner optimization; Deng *et al.* [11] revisits the optimization framework in [41] and observe that the inclusion of a momentum term in inner optimization can significantly enhance performance, leading to state-of-the-art results in certain settings.

DD is essentially a compression problem that emphasizes on maximizing the preservation of information contained in the data. We argue that well-defined metrics which measure the amount of shared information between variables in information theory are necessary for success measurement, but are

never considered by previous works. Therefore, we propose to introduce a well-defined metric in information theory, mutual information (MI), to guide the optimization of synthetic datasets.

**Contrastive Learning and Mutual Information.** The fundamental idea of all contrastive learning methods is to draw the representations of positive pairs closer and push those of negative pairs farther apart within a contrastive space. Several self-supervised learning methods are rooted in well-established ideas of MI maximization, such as Deep InfoMax [18], Contrastive Predictive Coding [29], MemoryBank [42], Augmented Multiscale DIM [2], MoCo [16] and SimSaim [8]. These are based on NCE [15] and InfoNCE [18] which can be seen as a lower bound on MI [30]. Meanwhile, Tian *et al.* [39] and Chen *et al.* [7] extend the contrastive concept into the realm of Knowledge Distillation (KD), pulling and pushing the representations of teacher and student.

The formulation of our method for DD, `MIM4DD` also absorbs the core idea (*i.e.*, constructing the informative positive and negative pairs for contrastive loss) of the existing contrastive learning methods, especially the contrastive KD methods, CRD [39] and WCoRD [7]. However, our approach has several differences from those methods: (i) our targeted MI and formulated numerical problem are totally different; (ii) our method can naturally avoid the cost of MemoryBank [42] for the exponential number of negative pairs in CRD and WCoRD, thanks to the small size of the synthetic dataset in our task. Given that the size of the synthetic dataset $M$ typically ranges from $0.1 - 1\%$ of the size of the real dataset $N$, the product $M \cdot N$ is significantly smaller than $N \cdot N$ (*i.e.*, $M \cdot N \ll N \cdot N$).

**Difference with DCC [26].** Recently, Lee et al. [26] introduced Dataset Condensation with Contrastive Signals (DCC), modifying the loss function to allow Dataset Condensation methods to effectively discern differences between classes. However, several distinctions exist between DCC and our method: **(i)** They are motivated differently. Our approach is predicated on information degradation, while DCC hinges on class diversity. **(ii)** From the perspective of contrastive learning, the view, positive and negative samples differ considerably. Our approach can be implemented at the feature map level, thanks to the introduced Theorem 1, while DCC can only be deployed at the gradient level. **(iii)** The performance of our method significantly surpasses that of DCC.

## C Codes

**Codes can be found anomalously in Supplement.**

