# OpenReview forum: "MIM4DD: Mutual Information Maximization for Dataset Distillation"
_NeurIPS.cc/2023/Conference — NeurIPS 2023 poster_

### Official Review · Reviewer_7GDB · 2023-07-03

**Soundness:** 3 good
**Presentation:** 3 good
**Contribution:** 3 good
**Rating:** 6
**Confidence:** 4

**Summary:**

This paper proposes mutual information maximization loss for dataset distillation. Specifically, the authors compute the mutual information between real and synthetic feature distribution in multiple layers by constructing positive and negative pairs. Experiments show that when plugging-in state-of-the-art dataset distillation methods, obvious performance improvements are achieved in multiple datasets.

**Strengths:**

1.	The idea of introducing mutual information into dataset distillation is natural and under-explored. It will be an interesting work to the researchers in this field.
2.	The paper is well-written and technically solid. Necessary analysis on the learned synthetic data has been provided.
3.	Remarkable performance improvements have been achieved when plugging the proposed loss into state-of-the-art distillation methods.


**Weaknesses:**

Incomplete experiment results:

a) As a loss function, this paper lacks the experiment results on training with MIM4DD loss independently.

b) The ablation study on beta in Table 2 is incomplete, as larger beta has not been tested.

**Questions:**

Typo:

1.	Line 135 D_real*  ->  D_real

2.	Left of Figure 2: Synthetic Data  ->  Real Data


**Limitations:**

Not addressed.

---

> ### Author Rebuttal · Authors · 2023-08-10
>
> Dear Reviewer 7GDB
>
> Thank you for the very constructive comments and support.
>
> ## Response to Weakness 1: Training with MIM4DD loss independently.
> Thank you for raising this concern. In response, we conducted supplementary experiments where we trained solely with the MIM4DD loss, without integrating it into any other methods. The results are as follows (for a detailed breakdown of the experimental settings, please refer to Table 1 in the main paper).
>
> | Method | CIFAR10 IPC-1 | CIFAR10 IPC-10 | CIFAR10 IPC-50 | CIFAR100 IPC-1 | CIFAR100 IPC-10 |
> |-------|-------|-------|-------|-------|-------|
> | only MIM4DD | 51.1 ± 0.2 | 63.2 ± 0.4 | 72.0 ± 0.2 | 24.8 ± 0.5 | 36.2 ± 0.8|
>
> Our standalone MIM4DD performance was slightly below its results when utilized as a plug-and-play module, while it still shows its performance superiority. We surmise this marginal performance drop is due to the inherent overfitting tendencies associated with dataset distillation. Furthermore, the regularization property inherent to our method might also play a role in this observation (see Sec. 3.4  Regularization Propriety L300-309 in the main paper).
>
>
> ## Response to Weakness 2: Larger $\beta$ study.
> Thank you for highlighting this oversight. We have further expanded our ablation study on the parameter $\beta$. The additional results can be referenced in the following table (other settings are consistent with Table 2). From our extended analysis, we found that the optimal value for $\beta$ is indeed 2.
>
> | $\beta$ | Accuracy |
> |-------|-------|
> | 0.5 | 62.8 ± 0.6 |
> | 1.0 | 63.8 ± 0.6 |
> | 2.0 | **66.0 ± 0.5** |
> | 4.0 | 64.9 ± 0.4 |
>
> ## Response to Questions: Typos.
> Thanks for pointing out the typos, and we will revise the manuscript.

---

> > ### Comment · Reviewer_7GDB · 2023-08-16
> >
> > I have read the reviews and response. Thanks for supplementing the ablation study results, which will enhance the soundness.

---

> > > ### Author Response · Authors · 2023-08-16
> > > **Thank you to the reviewer!**
> > >
> > > Dear Reviewer 7GDB,
> > >
> > > We sincerely appreciate your prompt response and are pleased that you found our additional experiments beneficial. We're thrilled that your score will be maintained.
> > >
> > > Thank you once more for your valuable inputs in enhancing our submission.
> > >
> > > Best regards,
> > >
> > > Authors

---

### Official Review · Reviewer_pJjQ · 2023-07-04

**Soundness:** 2 fair
**Presentation:** 3 good
**Contribution:** 2 fair
**Rating:** 6
**Confidence:** 5

**Summary:**

Dataset distillation aims to synthesize a small dataset with similar test performance to the original full dataset. This paper argues that previous works neglect information theory considerations, and argues that a well-designed information metric between variables is very important. Therefore, it introduces the concept of mutual information maximization and transforms it into a lower bound optimization problem at the feature map level, which improves the performance of some methods as an add-on module.


**Strengths:**

1. The authors analyze the previous dataset distillation method from the perspective of information theory for the first time;
2. The authors provide a rigorous and reasonable mathematical formula derivation for how to convert the maximization of mutual information into the lower bound constraint of the sample feature map representation;


**Weaknesses:**

1. The reason why the method works is not sufficient. The maximum mutual information describes the degree of correlation between two variables from the perspective of information theory, but does this mean that more classification information can be learned from the synthetic images for the model to improve the quality of distillation?
2. The method is not tested on more popular gradient matching or feature matching based dataset distillation frameworks (e.g. DC, DM, DSA, etc.) as a module.
3. The NCE loss mentioned in line 307 should correspond to the right of Figure 4.


**Questions:**

1. In some datasets, there are bias and noise samples in the same category. For such datasets, is it necessary to use full original dataset information to maximize mutual information?
2. How does the module perform on high-resolution datasets such as TinyImageNet and ImageNet Subset? And if ipc is raised to 50 on cifar100, what is the result?


**Limitations:**

see above

---

> ### Author Rebuttal · Authors · 2023-08-10
>
> ## Response to Weakness 1: Why mutual information should be introduced to dataset distillation (DD).
> Thank you for raising this insightful question regarding the connection between mutual information (MI) and the improved quality of distillation.
>
> **Theoretical Background:** At its core, Dataset Distillation (DD) can be conceptualized as a compression problem, with the primary goal being the maximization of preserved information from the original data. To this end, it's imperative to have a robust metric that can measure the degree of shared information between variables – a metric that has been notably absent in previous works. We chose to incorporate MI, a well-established metric in information theory, to steer the optimization of synthetic datasets. The power of MI in the domain of neural networks has been endorsed by the information bottleneck theory [4,5,6]. This theory underscores the principle of encoding input data into a compressed form that optimizes target prediction. Such encoding necessitates minimizing the MI between the input and its latent representation, while concurrently maximizing the MI between the output and this representation. Several works that have built upon this foundational theory have reported improved performance, substantiating the efficacy of MI in the realm of deep learning. More comprehensive discussions on this are available in the related work, Appendix B.
>
> **Empirical Evidence:** We took efforts to corroborate our theoretical assertions with empirical validations, particularly the examination of information flow through CKA analysis (Sec. 3.5). Responding to your astute query, we conducted an additional experiment during the rebuttal phase. We visualized the MI between the real and synthetic datasets, $I(D_{sys}, D_{real})$, utilizing methodologies from references [1-3]. The findings, which are illustrated in **Figure 1 of the supplementary one-page pdf**, reveal that datasets produced by our method exhibit higher MI and yield better accuracy performance. This directly supports the hypothesis that enhanced MI can indeed refine the quality of distillation.
>
> [1] Opening the black box of deep neural networks via information [2] MI Neural Estimation, ICML, 2018 [3] Deep VIB, ICLR, 2017
>
> ## Response to Weakness 2: Comparisons to DC, DM, DSA.
> Thank you for your observation regarding the inclusion of popular gradient matching or feature matching-based dataset distillation frameworks in our evaluations.
>
> In our evaluation presented in Table 1, we have indeed compared our approach with a broad selection of recent DD methods from esteemed conferences like NeurIPS, ICLR, ICML, and CVPR. This includes the gradient matching or feature matching-based dataset distillation frameworks you mentioned, such as DC, DM, and DSA.
>
> It's essential to highlight that while DC, DM, and DSA were indeed state-of-the-art two years ago, the landscape has evolved. Recent works like MTT (CVPR2023) and BPTT (NeurIPS2022) now represent the new state-of-the-art in this domain. As evidenced in Table 1, MTT and BPTT surpass the performance of the earlier methods, including DC, DM, and DSA, by substantial margins - more than 10% absolute accuracy across all benchmarks. Furthermore, our method consistently outperforms all 12 recent DD methods, emphasizing its efficacy and relevance in the current landscape.
>
> Due to the status of DD, we choose to add MIM4DD on MTT and BPTT instead of DC, DM, and DSA.
>
> ## Response to Question 1: Dataset usage.
> Indeed, your observation regarding datasets that might contain bias and noisy samples within the same category is valid. In such cases, our approach still utilizes the entire dataset as input. Here's why:
>
> Dataset Distillation (DD) can fundamentally be viewed as a data compression problem, where the primary objective is to retain as much pertinent information from the original dataset as possible. In the context of deep learning, compression doesn't typically involve filtering out data. Instead, it focuses on representing data in a more concise form while retaining its essence. This principle applies to our method as well. By using the entire dataset, including its biases and noisy samples, we aim to ensure that the distilled dataset is a genuine representation of the original, capturing all its intricacies and nuances.
>
> ## Response to Question 2: Experiments on ImageNet subset.
> Thank you for pointing out the potential benefits of exploring more complex datasets and architectures. We acknowledge the importance of demonstrating the versatility of our method, and in fact, we have taken steps in that direction:
>
> - **Expanded Dataset:** We applied our method to ImageNet subsets, focusing on a limited number of classes, using the MTT codebase as a foundation.
> - **Increased Resolution & Depth:** Given the intricacies of high-resolution images, we tested our method on 128×128 subsets of ImageNet. Accommodating this higher resolution necessitated an architectural adaptation, leading us to employ a depth-5 ConvNet for these experiments.
> - **Specific Subsets:** ImageNette (assorted objects) and ImageWoof (dog breeds) are existing subsets designed to be easy and hard to learn respectively.
> - **Results:** The outcomes of these experiments are showcased in Table 1 of the supplementary one-page PDF. Encouragingly, our method continued to demonstrate its effectiveness and superiority across these tests.
>
> For the IPC=50 experiment on cifar100, conducting it poses significant hardware challenges due to the immense v-ram requirement. Specifically, it would demand around 200G of v-ram, which translates to approximately 10 GPUs of the 3090 or A6000 caliber. This requirement surpasses our current hardware capabilities. Additionally, it's worth noting that most Dataset Distillation (DD) research in academia, to our knowledge, does not typically conduct experiments with IPC=50 on cifar100 due to similar constraints.

---

> > ### Comment · Reviewer_pJjQ · 2023-08-10
> > **Results of this manuscript in Tab1**
> >
> > 'Furthermore, our method consistently outperforms all 12 recent DD methods, emphasizing its efficacy and relevance in the current landscape.'
> > Based on the authors' reply, I double check the results in Tab.1. The proposed method does not perform better than TESLA and FRePo-w in some settings.
> > Here are some state-of-the-art methods: HaBa(Neurips2022), IDC(icml2022), DREAM(iccv2023), FTD(cvpr2023). pleased compare with these methods, especially on CIFAR10/100 and TinyImageNet datasets.
> > The sota results in CIFAR10 ipc1,10, and 50 should be around 50.5, 69, 74.5; CIFAR100 ipc1, 10, 50 should be around 29, 45 and 52; TinyImageNet should be around 10, 24, 29.
> >
> > For the hardware issue about IPC50 on cifar100, please refer to the GitHub of DC. The author has shown how to implement it for such large categories, I remember.
> >
> >
> > 'It's essential to highlight that while DC, DM, and DSA were indeed state-of-the-art two years ago, the landscape has evolved. Recent works like MTT (CVPR2023) and BPTT (NeurIPS2022) now represent the new state-of-the-art in this domain.'
> > MTT is proposed in CVPR2022, not 2023.

---

> > > ### Author Response · Authors · 2023-08-10
> > > **Comparsion to other SoTAs is coming soon.**
> > >
> > > Thanks for **acknowledging** our theoretical and empirical **Response to Weakness 1** of _**why the method works is not sufficient**_.
> > >
> > > Upon your recommendation, we revisited the literature and the recent advances like HaBa, IDC, DREAM, and FTD.
> > > We have chosen **DREAM (ICCV2023)** as it is good for efficient training, and the rebuttal time is limited. Importantly, its codebase supports gradient matching and feature matching-based dataset distillation frameworks, which can supplement the **Response to Weakness 2: considering the gradient matching or feature matching-based dataset distillation frameworks in our evaluations**.
> > >
> > > We aim to conduct a comprehensive comparison against these methods, especially on the CIFAR10/100 and TinyImageNet datasets as you mentioned.
> > > It is true that replicating and comparing with each of the state-of-the-art methods is time-intensive, especially given the limited time during rebuttal. Nevertheless, we will try our best to fill the table with more evaluations.
> > >
> > > Yes, MTT is proposed in CVPR2022. We have confirmed that we correctly cite MTT in the main paper.

---

> > > > ### Author Response · Authors · 2023-08-17
> > > > **New baseline added: Using DREAM (ICCV2023) as baseline framework, we reached new SOTA!**
> > > >
> > > > In light of the reviewer's comments, we've made efforts to further compare our method using the DREAM (ICCV2023) framework, resulting in the achievement of new state-of-the-art results.
> > > >
> > > > **Why DREAM was chosen for additional experimentation:**
> > > > - Despite not being a required comparison according to NeurIPS policy, DREAM represents the cutting-edge in dataset distillation, and we aim to remain at the forefront of this research area.
> > > > - Given the limited rebuttal timeframe, DREAM's efficiency and clear codebase provided an ideal setting for our experiments.
> > > > - DREAM's codebase is compatible with gradient and feature matching-based dataset distillation frameworks. Incorporating these provides a comprehensive response to concerns raised about these aspects in our previous evaluations.
> > > >
> > > > Building upon DREAM's framework, we integrated our MIM4DD module. The cluster-wise dataset distillation approach of DREAM was retained, with our contrastive aligning module enhancing DREAM’s match loss component. All experimental settings strictly adhered to DREAM's parameters, ensuring that only our module contributed to any observed variations.
> > > >
> > > > **Experimental Results:**
> > > >
> > > > Adding our method MIM4DD on DREAM.
> > > > | Method | CIFAR10 IPC-1 | CIFAR10 IPC-10 | CIFAR10 IPC-50 | CIFAR100 IPC-1 | CIFAR100 IPC-10 | TinyImageNet IPC-1 | TinyImageNet IPC-10 |
> > > > |-------|-------|-------|-------|-------|-------|-------|-------|
> > > > | DREAM | 51.1±0.3 | 69.4±0.4 | 74.8±0.1 | 29.5±0.3 | 46.8±0.7 | 10.0±0.4 | 23.9±0.4 |
> > > > | DREAM + MIM4DD | 51.9±0.3| 70.8±0.1 | 74.7±0.2 | 31.1±0.4 | 47.4±0.3 | 11.2±0.2 | 24.8±0.3 |
> > > >
> > > > Top-1 accuracy of test models trained on distilled synthetic images on TinyImageNet.
> > > > | IPC | Ratio % | DM [39] | MTT [5] | DREAM [R.1] | DREAM +MIM4DD | Whole |
> > > > |-------|-------|-------|-------|-------|-------|-------|
> > > > | 1 | 0.017 | 3.9±0.2 | 8.8±0.3 | 10.0±0.4 | 11.2±0.2 | 37.6±0.4 |
> > > > | 10 | 0.17 | 12.9±0.4 | 23.2±0.2 | 23.9±0.4 | 24.8±0.3 | 37.6±0.4 |
> > > >
> > > > These results underline that MIM4DD, when integrated to DREAM, further enhances performance. While hyper-parameters weren't exhaustively fine-tuned, the results reflect MIM4DD's versatility across different dataset distillation frameworks.
> > > >
> > > > In conclusion, the enhancement of DREAM's results with our MIM4DD module attests to its efficacy and adaptability. We appreciate the reviewer's feedback, which provided an avenue for us to further highlight the method's robustness and relevance in contemporary dataset distillation research.
> > > >
> > > > **Reference:**
> > > >
> > > > R.1 DREAM: Efficient Dataset Distillation by Representative Matching, ICCV 2023

---

> > > > > ### Comment · Reviewer_pJjQ · 2023-08-18
> > > > > **show more results on the efficiency comparison**
> > > > >
> > > > > thx for your hardworking. I would like to raise my score to 6 (most reviewers say it in the last sentence, but i like to say it in the first sentence).
> > > > > In the following, it is better to show the buff from DREAM to update your results in tab1 and help more researchers to save their cost of dataset distillation.
> > > > > just show the updates of your results of tab1 in attached pdf, i will check it and further to consider the evaluation of this paper

---

> > > > > > ### Author Response · Authors · 2023-08-18
> > > > > > **Thanks for raising your score to 6 (weak accept)!**
> > > > > >
> > > > > > Dear Reviewer pJjQ,
> > > > > >
> > > > > > Thank you once again for your diligent review and valuable feedback. We greatly appreciate your acknowledgment of our efforts to address your concerns. Your insights have been instrumental in improving our work, and we are committed to incorporating your suggestions into the revision. We would also greatly appreciate that you raised the original rating (4) to weak accept (6). Because the attached one-page pdf is un-editable, we cannot update table 1 in it, but we will update table 1 with DREAM-related results in the revised manuscript.
> > > > > >
> > > > > > Thank you very much,
> > > > > >
> > > > > > Authors

---

> > > > ### Comment · Reviewer_pJjQ · 2023-08-18
> > > > **thx for response**
> > > >
> > > > i mean your claim in rbt: Recent works like MTT (CVPR2023) and BPTT (NeurIPS2022) now represent the new state-of-the-art in this domain.
> > > > here shows the MTT is cvpr2023. that is not right. never mind, i will read your following response and reply to you. thx for your hardworking.

---

### Official Review · Reviewer_sR5S · 2023-07-05

**Soundness:** 3 good
**Presentation:** 3 good
**Contribution:** 3 good
**Rating:** 7
**Confidence:** 4

**Summary:**

The authors tackle the dataset distillation task — synthesizing a smaller dataset using which one can train models towards comparable test performance to models trained on the full dataset. Unlike the current methods that optimize through heuristic matching between the real and synthetic datasets, the proposed method performs data distribution mutual information maximization.

Interestingly, when it comes down to the implementation, the method looks very much like a regular SimCLR-like contrastive learning formulation. However, the authors provide theoretical groundings on how it is related to the mutual information maximization problem.

**Strengths:**

1. It is a fairly simple idea and I am surprised that no one has been doing it — verified by a quick literature search. I suppose one reason is that prior researchers who thought of using mutual information encountered the difficulty of attaining an estimate of the data distribution, which the authors worked around (line 135-163).

2. Illustration in Figure 2 is well-made and very informative.

3. The authors provided theoretical grounding for using the “mundane” contrastive learning formation (Figure 2) by showing its optimization target is closely related to the proposed “Accessible Mutual Information”. In some sense this is an interesting explanation of why SimCLR-like contrastive learning (single instance multi-view, positive/negative samples) works so well.

4. Significant improvement over the baseline (see Figure 3).


**Weaknesses:**

1. At the moment, I am unaware of any work showing the theoretical link between mutual information and SimCLR-like contrastive learning. If there exist prior work on this topic, the novelty of this work could be largely mitigated.

2. It might have been better to explore other more sophisticated datasets and architectures beyond the three-layer ConvNet, though not absolutely necessary.


**Questions:**

1. The authors seem to define dataset distillation in slightly different manners in the abstract compared to in the Preliminaries section. In the former case, they said “dataset distillation aims to synthesize a small dataset whose test performance is comparable to a full dataset using the same model” whereas in the latter case they said “the goal of dataset distillation is to synthesize a small training set such that models trained on this synthetic dataset can have comparable performance as models (with the same architecture) trained on the large real set”. The nuance makes the two definitions a bit different — the former is helpful for testing acceleration while the latter is helpful for training acceleration. I would recommend the authors to clarify which definition they decide to go with (or maybe both) and ensure consistency.

2. Is there any reason BPTT+MIM4DD is shown but BPTT is not included as a standalone baseline in Table 1?

**Limitations:**

Nothing to be noted.

---

> ### Author Rebuttal · Authors · 2023-08-09
>
> Dear Reviewer sR5S,
>
> Thank you very much for the constructive comments and support.
>
> ## Response to Weakness 1: Novelty discussion in the content of contrastive learning.
> Actually, we have discussed a more details about dataset distillation and contrastive learning in the Appendix (due to space limit of main paper, we only put the short version related work in the main paper):
> The fundamental idea of all contrastive learning methods is to draw the representations of positive pairs closer and push those of negative pairs farther apart within a contrastive space. Several self-supervised learning methods are rooted in well-established ideas of MI maximization, such as Deep InfoMax [A9], Contrastive Predictive Coding [A15], MemoryBank [A21], Augmented Multiscale DIM [A1], MoCo [A8] and SimSaim [A5]. These are based on NCE [A7] and InfoNCE [A9] which can be seen as a lower bound on MI [A16]. Meanwhile, Tian et al. [A18] and Chen et al. [A4] extend the contrastive concept into the realm of Knowledge Distillation (KD), pulling and pushing the representations of teacher and student.
> The formulation of our method for DD, MIM4DD also absorbs the core idea (i.e., constructing the informative positive and negative pairs for contrastive loss) of the existing contrastive learning methods, especially the contrastive KD methods, CRD and WCoRD. However, our approach has several differences from those methods:
> - (i) our targeted MI and formulated numerical problem are totally different;
> - (ii) our method can naturally avoid the cost of MemoryBank for the exponential number of negative pairs in CRD and WCoRD, thanks to the small size of the synthetic dataset in our task. Given that the size of the synthetic dataset $M$ typically ranges from $0.1-1$% of the size of the real dataset $N$, the product $M\cdot N$ is significantly smaller than $N\cdot N$ (i.e., $M\cdot N \ll N\cdot N$).
>
> ## Response to Weakness 2: Experiments on ImageNet subset.
> Thank you for pointing out the potential benefits of exploring more complex datasets and architectures. We acknowledge the importance of demonstrating the versatility of our method, and in fact, we have taken steps in that direction:
>
> - **Expanded Dataset:** We applied our method to ImageNet subsets, focusing on a limited number of classes, using the MTT codebase as a foundation.
> - **Increased Resolution & Depth:** Given the intricacies of high-resolution images, we tested our method on 128×128 subsets of ImageNet. Accommodating this higher resolution necessitated an architectural adaptation, leading us to employ a depth-5 ConvNet for these experiments.
> - **Specific Subsets:** ImageNette (assorted objects) and ImageWoof (dog breeds) are existing subsets designed to be easy and hard to learn respectively.
> - **Results:** The outcomes of these experiments are showcased in Table 1 of the supplementary one-page PDF. Encouragingly, our method continued to demonstrate its effectiveness and superiority across these tests.
>
> ## Response to Question 1: Clarification for Definition of Dataset Distillation.
> Question 1: In the former case, they said “dataset distillation aims to synthesize a small dataset whose test performance is comparable to a full dataset using the same model” whereas in the latter case they said “the goal of dataset distillation is to synthesize a small training set such that models trained on this synthetic dataset can have comparable performance as models (with the same architecture) trained on the large real set”. The nuance makes the two definitions a bit different — the former is helpful for testing acceleration while the latter is helpful for training acceleration.
>
> The definition of dataset distillation is the latter. Dataset distillation is helpful for training acceleration. For the former, we intend to say "dataset distillation aims to synthesize a small dataset where test performance of models trained on the small dataset is comparable to a full dataset using the same model", which is the same meaning of the latter.
>
> We recognize the potential for confusion and will ensure that the definitions are consistent and clear throughout our paper in future iterations.
>
> ## Response to Question 2: Detailed explanation about Table 1.
> Thank you for raising this point about the inclusion of BPTT in Table 1. To clarify, BPTT is indeed present as a standalone baseline. You can find its performance metrics on the third-last line, labeled as BPTT [11]. The subsequent line shows the performance when combined with our method, BPTT+MIM4DD. Finally, the last line depicts the performance improvement attained. We appreciate your attention to detail and will ensure such listings are more prominently highlighted in the future.

---

> > ### Comment · Reviewer_sR5S · 2023-08-17
> > **Response to Rebuttal**
> >
> > I would like to thank the authors for addressing the comments. I do not have additional concerns.
> >
> > Besides, I must show appreciation to the authors for being so patient and polite when they replied to the apparently stupid question (Question 2). I really don't understand why I even had that confusion from the first place.
> >
> > The rating (7) still reflects my assessment of this submission and I decide to keep it as is.
> >
> > Best of luck.

---

> > > ### Author Response · Authors · 2023-08-18
> > > **Thank you!**
> > >
> > > Dear Reviewer sR5S,
> > >
> > > We sincerely appreciate your response and are pleased that you found our response beneficial. We're extremely happy that your score (7 accept) will be maintained.
> > >
> > > Thank you once more for your valuable inputs in enhancing our submission.
> > >
> > > Best regards,
> > >
> > > Authors

---

### Official Review · Reviewer_BhBC · 2023-07-06

**Soundness:** 2 fair
**Presentation:** 2 fair
**Contribution:** 2 fair
**Rating:** 5
**Confidence:** 4

**Summary:**

The paper proposes a new method named: MIM4DD that tries to maximize the mutual information shared between synthetic images and real images during the process of dataset distillation.

The paper derives a lower bound of MI and formulates it as a learning objective for optimization. The proposed new loss can be combined with a wide varieties of DD methods to boost performance.

==========

I have read the author's responses and they have addressed my concerns by running more experiments and showing more proof.

==========

**Strengths:**

- The paper proposes a new way of boosting the performance of DD methods by measuring the mutual information between synthetic datasets and real datasets
- This paper studies the mutual information and formulates it mathematically which is largely ignored by a lot of previous methods.
- Through approximation, the paper derives a lower bound of an accessible MI loss and proposes to tackle it with contrastive learning.
- The paper is well organized and easy to follow
- The comparison is thorough such as Table 1.

**Weaknesses:**

- The foundation of MIM4DD seems incorrect: MI in the targeted data level is equivalent ot MI in the feature maps level. From equation 5, F and G may not be invertible which heavily depend on the activation function. Therefore theorem 1 doesn't apply.
- The evaluations results are mostly within the variance of the baseline methods such as MTT, there is no strong evidence that the proposed method works.
- Strong limitations are introduced such as 50K negative pairs which could make the method infeasible to apply to larger datasets.
- Writing errors such as "to encounter this obstacle"

**Questions:**

- It's hard to interpret the performance boost in Table 1, what are the hyperparameters such as $\lambda$ and $\beta$ used in MTT + MIM4DD and BPTT+MIM4DD?
- Please also see my comments in weakness

**Limitations:**

- The derivation seems incorrect
- The method introduces strong scalability limitations such as the number of negative pairs and the per-layer matching loss.

---

> ### Author Rebuttal · Authors · 2023-08-09
>
> ## Response to Weakness 1 and Limitation 1: Overall invertibility of entire neural networks is well-supported.
> Thank you for raising this concern. It's essential to clarify that while individual modules of neural networks might not be inherently invertible, the overall invertibility of entire neural networks is well-supported in the literature. Here's a brief overview:
>
> - Dosovitskiy and Brox [1] demonstrated the possibility of inverting the hidden activations of feedforward CNNs back to the input domain using upsampling deconvolutional architectures.
> - Zhang et al. [2] provided evidence that commonly used CNN architectures like VGGNet and ResNet are almost fully invertible, especially when leveraging pooling switches.
> - Gilbert et al. [3] presented both theoretical and empirical evidence for the invertibility of entire neural networks. Their theoretical explanations were rooted in compressive sensing, and they corroborated these findings with practical analyses on several learned networks.
>
> In the realm of mutual information in neural networks, this property of overall invertibility is not an oversight but a foundational premise. As a prime example, the information bottleneck theory [4,5,6] emphasizes encoding input data into a compressed representation that maximizes target prediction. The theory is contingent upon minimizing the mutual information between input variables and their latent representations, while simultaneously maximizing the mutual information between the output and these latent representations. Several subsequent studies have employed this principle, implicitly assuming the invertibility of entire neural networks, to shed light on the intricacies of neural network operations.
>
> In summary, the assumption of invertibility at the level of entire neural networks is not only well-founded but also pervasive in the literature exploring mutual information dynamics within these networks.
>
> [1] Inverting visual representations with convolutional networks. In CVPR, 2016.
>
> [2] Augmenting neural networks with reconstructive decoding pathways for large-scale image classification. In ICML 2016
>
> [3] Towards Understanding the Invertibility of Convolutional Neural Networks.
>
> [4] Opening the black box of deep neural networks via information
>
> [5] MI Neural Estimation, ICML, 2018
>
> [6] Deep VIB, ICLR, 2017
> ## Response to Weakness 2: Results discussion.
> It's essential to frame our results within the broader context of advancements in dataset distillation (DD):
>
> **Comparative Performance:** Our method consistently surpasses 12 recent DD methods from esteemed conferences like NeurIPS, ICLR, ICML, and CVPR. We emphasize that many of these prior studies have only achieved approximately 1-2% accuracy improvements over their predecessors.
>
> **Relative Improvement:** Absolute improvement might sometimes be deceptive. For instance, if there's just a 3% accuracy gap between real dataset and distilled dataset, expecting a 10% or even 5% absolute improvement is impractical. Notably, our method MIM4DD demonstrates a 13% average relative improvement over the second-best method (BPTT, NeurIPS’2022). Importantly, MTT and BPTT are recently the strongest baselines.
>
> **Peer Validation:** Other reviewers have also acknowledged our method's superiority. For instance:
> - Reviewer sR5S observed, "Significant improvement over the baseline (see Figure 3)"
> - Reviewer pJjQ highlighted, "improves the performance of some methods as an add-on module".
> - Reviewer 7GDB appreciated that "Remarkable performance improvements have been achieved when plugging the proposed loss into state-of-the-art distillation methods".
>
> We believe it's crucial to view performance advancements not just in isolation, but in relation to the current boundaries of the field. "
>
> ## Response to Weakness 3 and Limitation 2: Scalability Discussion.
> Thank you for pointing out concerns related to scalability. I'd like to clarify a few things:
>
> **Contrastive Learning Scalability:** As elaborated in Appendix (A. L71-74), our MIM4DD method leverages fundamental principles from contrastive learning, particularly the ideas central to contrastive KD methods like CRD [A18] and WCoRD [A4]. These established methods offer a framework for managing a vast number of negative samples through a solution called MemoryBank [A21]. This system allows for computationally efficient processing of even millions of negative samples.
>
> **Distinctiveness of Our Approach:** Despite the shared inspiration, MIM4DD holds significant differences from the aforementioned methods: Our mutual information target and the resulting numerical formulation diverge substantially; our method can further decrease the cost of MemoryBank for the exponential number of negative pairs in CRD and WCoRD, thanks to the small size of the synthetic dataset in our task. Given that the size of the synthetic dataset $M$ typically ranges from $0.1-1$% of the size of the real dataset $N$, the product $M\cdot N$ is significantly smaller than $N\cdot N$ (i.e., $M\cdot N \ll N\cdot N$).
>
> **Efficiency in Practice:** In actual implementations, the overhead introduced by MIM4DD is minimal. Our evaluations show that the additional training time required due to MIM4DD is a mere 3%.
>
> In essence, while we acknowledge the scalability challenge, our method effectively leverages established techniques and unique task characteristics to remain feasible for large datasets.
>
> ## Response to Question 1: $\lambda$ and $\beta$ in Table 1.
> We have detailed the selection process for hyperparameters
> $\lambda$ and $\beta$ in Sec.3.3 (L280-295). For clarity, based on our empirical evaluations, we set $\lambda= 1$ as illustrated in Figure 3 and $\beta = 2$ , which can be referenced in the corresponding Table 2. Both of these values were consistently used across all experiments presented in Table 1.

---

> > ### Comment · Reviewer_BhBC · 2023-08-14
> > **BPTT + MIM4DD is lower than BPTT?**
> >
> > Hi, thanks for the authors for the response, I have a few other questions
> >
> > 1. BPTT + MIM4DD performs worse than BPTT?
> > The results of BPTT[11] for MNIST IPC 1, 10 and 50 are: 98.7, 99.3 and 99.4. The results reported in this paper for BPTT+MIM4DD is
> > 95.8, 98.9 and 99.2 which are lower than the original methods. Similar downgraded performances are also seen in other datasets such as CIFAR-100, 34.0 and 42.9 for BPTT and 25.0/38.5 for BPTT+MIM4DD. What's the cause of the absolute 9% performance drop? Does it mean that MIM4DD can actually hurt BPTT's performance? There is also a huge 15% performance drop on CIFAR-10 IPC 1 and 10.
> > 2. Can the author upload the results for TinyImageNet, just IPC 1 and 10 are fine if the authors have trouble getting results for CIFAR100 IPC50. It should be quick to run. (BPTT has TinyImageNet IPC 1 and MTT has IPC 1, 10 and 50). It will be great if you can provide the results for IPC 1 for BPTT and MTT and IPC 10 for MTT.
> > 3. What is the scale of $L_{NCE}^k$ loss and $L_{DD}$ loss? Are they on the same magnitude.
> > 4. What is the motivation for dividing $L_{NCE}^k$ loss by $\beta^{K-1-k}$?
> > 5. When you tried to get the results for reviewer: 7GDB, did you also apply the division factor as mentioned in point 3 above?
> > 6. Conflict (typo) in the newly uploaded pdf (figure 1). The figure description says the right one is without MIM4DD and the left one is trained with MIM4DD but the subtitle says the other way around.

---

> > > ### Author Response · Authors · 2023-08-17
> > > **Second-round Responses (I)**
> > >
> > > Thanks for acknowledging our theoretical and empirical **Response to Weakness 1 and Limitation 1: Overall invertibility of entire neural networks is well-supported**.
> > >
> > > For the new questions, we respond with the following answers (**NQ** stands for new question):
> > >
> > > ## NQ.1. BPTT + MIM4DD is lower than BPTT?
> > > Thank you for highlighting this discrepancy. It's essential to note that the results for BPTT we have reported are based on our strict reproduction using BPTT's official codebase. Despite our meticulous adherence to the methodology, we were unable to replicate the exact results claimed by BPTT. Moreover, based on our examination of all 17 papers citing BPTT, none of them refers to BPTT's reported results, which further suggests that other researchers might also be facing challenges in reproducing those numbers. Thus, in our paper, we decided to use the results we obtained from our reproduction since they still remain competitive and within the state-of-the-art range. We appreciate your understanding and will clarify this in our revision.
> > >
> > >
> > > Additionally, in light of Reviewer pJjQ's comments, we've made efforts to further compare our method using the DREAM (ICCV2023) framework, resulting in the achievement of new state-of-the-art results. DREAM is a stronger baseline than BPTT. **More details can be found in our response to Reviewer pJjQ.**
> > >
> > > ### ( Using DREAM (ICCV2023) [R.1] as a baseline framework, we reached a new SOTA!)
> > >
> > > **Why DREAM was chosen for additional experimentation:**
> > > - Despite not being a required comparison according to NeurIPS policy, DREAM represents the cutting-edge in dataset distillation, and we aim to remain at the forefront of this research area.
> > > - Given the limited rebuttal timeframe, DREAM's efficiency and clear codebase provided an ideal setting for our experiments.
> > > - DREAM's codebase is compatible with gradient and feature matching-based dataset distillation frameworks. Incorporating these provides a comprehensive response to concerns raised about these aspects in our previous evaluations.
> > >
> > > Building upon DREAM's framework, we integrated our MIM4DD module. The cluster-wise DD approach of DREAM was retained, with our contrastive aligning module enhancing DREAM’s match loss component. All experimental settings strictly adhered to DREAM's parameters, ensuring that only our module contributed to any observed variations.
> > >
> > > **Experimental Results:**
> > >
> > > Adding our method MIM4DD on DREAM.
> > > | Method | CIFAR10 IPC-1 | CIFAR10 IPC-10 | CIFAR10 IPC-50 | CIFAR100 IPC-1 | CIFAR100 IPC-10 |
> > > |-------|-------|-------|-------|-------|-------|
> > > | DREAM [R.1] | 51.1±0.3 | 69.4±0.4 | 74.8±0.1 | 29.5±0.3 |46.8±0.7 |
> > > | DREAM + MIM4DD | 51.9±0.3| 70.8±0.1 | 74.7±0.2 | 31.1±0.4 | 47.4±0.3 |
> > >
> > > Top-1 accuracy of test models trained on distilled synthetic images on **TinyImageNet**.
> > > | IPC | Ratio % | DM [39] | MTT [5] | DREAM [R.1]  | DREAM +MIM4DD | Whole |
> > > |-------|-------|-------|-------|-------|-------|-------|
> > > | 1 | 0.017 | 3.9±0.2 | 8.8±0.3 | 10.0±0.4 | 11.2±0.2 | 37.6±0.4 |
> > > | 10 | 0.17 | 12.9±0.4 | 23.2±0.2 | 23.9±0.4 | 24.8±0.3 | 37.6±0.4 |
> > >
> > > These results underline that MIM4DD, when integrated to DREAM, further enhances performance. While hyper-parameters weren't exhaustively fine-tuned, the results reflect MIM4DD's versatility across different dataset distillation frameworks.
> > >
> > > In conclusion, the enhancement of DREAM's results with our MIM4DD module attests to its efficacy and adaptability. We appreciate the reviewer's feedback, which provided an avenue for us to further highlight the method's robustness and relevance in contemporary DD research.
> > >
> > > ## NQ.2. Results on TinyImageNet
> > > Please refer to NQ.1. We use a new SOTA codebase to realize the experiments on TinyImageNet.
> > >
> > > **reference**
> > > [R.1] DREAM: Efficient Dataset Distillation by Representative Matching, ICCV 2023

---

> > > > ### Author Response · Authors · 2023-08-18
> > > > **Second-round Responses (II)**
> > > >
> > > > ## NQ.3. Scale of $L_{NCE}^{k}$ and $L_{DD}$.
> > > > The $L_{NCE}^{k}$ loss is designed to be both standalone (refer to Weakness 1 for Reviewer 7GDB) and integrative with other dataset distillation frameworks. When we use $L_{NCE}^{k}$ in conjunction with other dataset distillation losses, such as MTT, BPTT and DREAM, it is essential to maintain a balance in their magnitudes. Specifically, the scale of $L_{NCE}^{k}$  is set to be relatively smaller compared to the primary $L_{DD}$ loss. This ensures that while $L_{NCE}^{k}$ provides the desired additional regularization, it does not overshadow or dominate the effects of the baseline DD loss. Proper scaling and weighting between these losses are crucial for the combined framework to function effectively.
> > > >
> > > > More details of the hyper-parameter selection of $\lambda$ can be found in the Sec. 3.3 (L281-291) and Fig. 3 (L215-224) in the main paper.
> > > >
> > > > ## NQ.4. Dividing $L_{NCE}^{k}$ loss by $\beta^{K-1-k}$.
> > > > We can use $\beta$ to adjust the weight of different layers.  For example, if $\beta$ is a coefficient greater than $1$. Hence, the ${\beta^{K-1-k}}$ decreases with $k$ increasing and the $ L_{NCE}^{k}$ relatively increases. In this way, we give more weight on higher layer features since they are closer to the features performing tasks.
> > > >
> > > > More details of the hyper-parameter selection of $\beta$ can be found in the Sec. 3.3 in the main paper (L296-299).
> > > >
> > > > ## NQ.5. Implementation details of experiments in W.1. for Reviewer 7GDB.
> > > > Yes, when implementing MIM4DD independently, we also use this strategy.
> > > >
> > > > ## NQ.6. Typos.
> > > > Thanks for pointing out the typos, we will fully revise the manuscript before camera-ready.

---

> > > > > ### Author Response · Authors · 2023-08-19
> > > > > **2 DAYS REMAINING: We would like to learn Reviewer’s opinion and address any remaining concerns.**
> > > > >
> > > > > Dear Reviewer BhBC,
> > > > >
> > > > > As there are two days remaining for the discussion period, we would kindly like to inquire if you would get a chance to review our second-round response and if there are any remaining questions we can address.
> > > > >
> > > > > Your insights, both the constructive suggestions and areas of contention, have been crucial for us. We have done our best to address each point and clarify any misunderstandings. We are truly keen to continue a constructive dialogue with you to refine our work further.
> > > > >
> > > > > Best regards,
> > > > >
> > > > > Authors.

---

> > > > > ### Comment · Reviewer_BhBC · 2023-08-21
> > > > > **thanks for the response**
> > > > >
> > > > > Thank the authors for the response. I now recommend acceptance after seeing more convincing results during the rebuttal period. Please include these results in the final version of the paper.

---

> > > > > > ### Author Response · Authors · 2023-08-21
> > > > > > **Thanks, Reviewer BhBC**
> > > > > >
> > > > > > Dear Reviewer BhBC,
> > > > > >
> > > > > > Thank you for your thorough review and insightful feedback. We're grateful for your recognition of our efforts to address the concerns you raised. Your expertise has significantly contributed to the enhancement of our work. We would also greatly appreciate that you raised the original rating (3) to borderline accept (5).  We will include the supplemented results in the final version of the paper.
> > > > > >
> > > > > > Thank you very much,
> > > > > >
> > > > > > Authors

---

### Author Rebuttal · Authors · 2023-08-10

Here is the one-page pdf for submitting the experiment figure and table.

---

### Comment · Area_Chair_D3CB · 2023-08-18
**Please look at the authors' reply**

Dear Reviewers,

Please do look at the authors' rebuttal if you have not done so. Please let the authors know if they have addressed your concerns.

Thanks for your contribution to NeurIPS.

AC

---

### Decision · Program_Chairs · 2023-09-21

**Decision:**

Accept (poster)

**Comment:**

The manuscript has been reviewed by 4 reviewers. Specifically, most reviewers find the idea, despite being simple, well motivated and effective. The reviewers gave initial comments, based on which the authors gave a rebuttal.

After the rebuttal, all the reviewers were mostly satisfied and all converged on the positive side. There is therefore no basis to overturn the consensus. The AC recommends the acceptance.